# Cycling Infrastructure for All EPACs Included?

Nikolaas Van den Steen [1,2,*], Bas de Geus [3,4], Jan Cappelle [1] and Lieselot Vanhaverbeke [2,*]

1   Electa Gent, KU Leuven Technology Campus, Gebroeders de Smetstraat 1, 9000 Ghent, Belgium; jan.cappelle@kuleuven.be
2   MOBI Research Group, Vrije Universiteit Brussel, Pleinlaan 2, 1050 Brussel, Belgium
3   Institute for the Analysis of Change in Historical and Contemporary Societies (IACCHOS), Université catholique de Louvain, Place Pierre de Courbertin, 1348 Louvain-la-Neuve, Belgium; bas.de.geus@uclouvain.be
4   Human Physiology and Sport Physiotherapy Research Group, Vrije Universiteit Brussel, Pleinlaan 2, 1050 Brussel, Belgium
*   Correspondence: nikolaas.vandensteen@kuleuven.be (N.V.d.S.); lieselot.vanhaverbeke@vub.be (L.V.)

**Abstract:** A modal shift to electric pedal-assisted cycles (EPACs) can help with reaching the transport emission goals of the European Green Deal. With the rising sales of EPACs in Europe, a lack of appropriate (electric) cycling infrastructure remains a major barrier for many potential users. This paper discusses the results of a survey about the requirements of (potential) cyclists to design a better cycling infrastructure. The differences in requirements for non-cyclists vs. cyclists and electric cyclists vs. conventional cyclists are discussed using statistical analysis. The key findings are that cyclists and non-cyclists both require wide quality cycling infrastructure with safe crossing points, secure bicycle parking and smart traffic lights. Non-cyclists' requirements significantly differ from cyclists' on 12 items, of which rain cover while cycling and parking spots for the car are the most noteworthy. There is (but) one significant difference between the requirements of EPAC users and conventional cyclists: the need for charging points for EPACs along the cycle route.

**Keywords:** bicycle; e-bike; infrastructure; light electric vehicle

## 1. Introduction

The Green Deal states that the European Union has to decrease the greenhouse gas (GHG) emissions caused by transport (25% of all GHG emissions) with 90% by 2050 [1]. A modal shift in transport, i.e., a shift to a sustainable and active means of transport instead of internal combustion engine vehicles (ICEVs), could contribute hugely to this objective. A 15 km ICEV commute replaced by an electric bicycle would save 2.8 kg $CO_2$eq. each commute [2]. Electric pedal-assisted cycles (EPACs) and light electric vehicles (LEVs) in general might provide an alternative to ICEVs [3–5].

### 1.1. Context

Already today, EPACs are taking an increasingly important position in the growing European bicycle market. Estimates from three European cycling associations (CIE, ECF and CONEBI [6–8]) state that over 17 million e-bikes will be sold by 2030 in Europe, a growth of 360% compared to 3.7 million e-bikes sold in 2019 [9] and from 2026 EPAC sales will be greater than conventional bicycles. In Belgium, recent numbers of Traxio (association of mobility partners) indicate that in 2019 49.16% of the sales in bicycle shops were electric bikes [10].

In Flanders, the northern part of Belgium, in addition to pedelecs (EPACs with pedal assist up to 25 km/h and max. 250 W continuous-rated motor power), speed pedelecs (EPACs with pedal assist up to 45 km/h and max. 4 kW continuous-rated motor power) are gaining popularity and see a steady incline in sales each year (with a small decline of 6.2% in 2020 due to the COVID-19 pandemic) [10]. Its use as a commuting vehicle is also on

the rise, with 5.2% of Flemish commuters in 2019 taking an EPAC to work [11], compared to 3.9% in 2018 [12] and 2.3% in 2017 [13]. However, with 84% of Flemish commuters living less than 30 km from their work (a feasible distance with a speed pedelec [14], there is still a margin to increase the number of conventional bicycle and EPAC commuters (total of 17% in 2019) [15] in order to contribute to the Green Deal objectives.

*1.2. Literature*

To further increase the number of cyclists, designing and building safe and adapted cycling infrastructure is necessary. The lack of appropriate infrastructure for cycling is a barrier for people to take up cycling. The literature clearly demonstrates that building cycling infrastructure has a positive impact on the modal share of both conventional and electric cycling. If quality cycling infrastructure is built, people will cycle on it [5,16–24]. In the case of electric cycling, the project 365SNEL learnt that e-cyclists are prepared to make a detour to use better infrastructure instead of the shortest or fastest route. Policymakers should be advised to increase investments in cycling infrastructure, as it is an advantageous investment for society, with the long-term health effects that are more beneficial than the cost of construction, maintenance and accidents [25,26]. Rich et al. even states the following on the effects of the large share of e-bikes: "*At the specific level, it is found that larger shares of e-bikes implies lower benefits as these bikes provide lower health benefits and larger accident costs*".

Several review papers on cycling infrastructure have been written over the years: ranging from the review paper of Heinen et al. [20], which discusses that infrastructure is one of the key determinants of commuting to work with a bicycle; the review article of Pucher et al. [27] that gives an overview of possible bicycle infrastructure; programs and policies to promote cycling; the review of DiGioia et al. [28] that differentiates cycling infrastructure in terms of safety; and the review by Buehler et al. [29] that discusses literature on bicycle parking. However, these reviews do not mention EPACs when discussing cycling infrastructure.

Research on motivations and barriers has shown similarities and differences in barriers between conventional cyclists (CC) and EPAC users, with 'bad weather' and especially rain being some of the biggest common barriers, besides the lack of infrastructure to take up cycling [30–32]. Even the introduction of quality cycling infrastructure would not increase cycling numbers on rainy days [24], this proved to be the case especially for non-cyclists. One study demonstrated that barriers between non-cyclists and cyclists, before they took up cycling, do not differ in low cycling countries [33]. However, the triggers that made cyclists start cycling do differ with the needs non-cyclists have when considering starting cycling. Research in Flanders and Brussels, in the capital region (Belgium), demonstrated that cycling infrastructure is essential, and when present, psycho-social determinants are more important than environmental factors [34–36].

What quality cycling infrastructure entails is detailed for Flanders in the '*Fietsvademecum*' [37]. This document was set up by the Flemish Government to be a guideline for quality control in the realization of the cycle route network and described appropriate cycling facilities, with guidelines on concrete design (dimensions, use of materials, etc.). On a higher level, the European Commission has basic quality design principles for cycle infrastructure and networks. The basic principles to be adhered to when designing and implementing cycle infrastructure are: Safety, directness, coherence, attractiveness, and comfort [38]. The literature also provides a means of evaluating the 'appropriateness' of existing infrastructure [39,40].

To the best of the knowledge of the researchers, there is no research into the difference of the needs of EPAC users for cycling infrastructure compared to the needs of conventional cyclists. It seems that the needs of conventional cyclists and EPAC users are assumed similar. However, the heavier weight [41], presence of an electric motor and battery, higher purchase price [30,31] and different speeds and accelerations [41–43] suggests the differences in needs. One Danish study, published after the set-up of this research, models the impact of cycle highways and EPACs and takes into account the rising share of EPACs [44]. In their model,

Hallberg et al. only makes a distinction between conventional bicycles, pedelecs and speed pedelecs based on speed. The preferred route choice is assumed to be based on minimizing the travel time. This is contrary to findings in the 365SNEL project [30], as explained above.

In conclusion, one of the essential barriers keeping the Flemish commuters from taking their (electric) bicycle to work is the lack of appropriate infrastructure [31,34–36]. The literature clearly shows that providing appropriate cycling infrastructure will increase cycling [5,16–24] and provides a means of evaluating the 'appropriateness' of existing infrastructure [27,37–40]. How future cycling infrastructure should look like from a (EPAC) user and potential (EPAC) user's point of view is yet unclear.

*1.3. Research Gap*

Considering the growing number in EPAC sales and the increased use as a commuting vehicle, one might not consider an EPAC as a 'new phenomenon' anymore on the existing cycling infrastructure. It raises the question whether the existing infrastructure is adequate for the (future) growing numbers of (electric) cyclists and if not, should the infrastructure be adapted to the needs of a growing number of EPAC users. With higher average and top speeds, higher purchase prices, wider handlebars and a battery on board, EPAC users' requirements in terms of cycling infrastructure are most likely different. To the best of our knowledge and at the time of the study, no research exists on the differences of needs between EPAC users and non-EPAC users (conventional cyclists) with regards to cycling infrastructure. This paper addresses this research gap by answering the following research questions:

- What kind of cycling infrastructure should be provided according to different types of cyclists?
- Are the requirements different according to cyclists and non-cyclists?
- Are the requirements different according to EPAC users and conventional cyclists?

## 2. Materials and Methods

This study was set up in the context of the Smart Energy Bike Path (SEBP) project, a Flemish FLUX50 project (start in 2019, end in 2021), in which a number of private and public partners wanted to investigate the feasibility of getting at least 10% of Flemish car commuters on a bicycle by offering a "*smart bike path solution*". The "*smart bike path*" would be a cycling path covered with solar panels, which would not only be providing a comfortable experience for cyclists being protected from weather, the power necessary for smart technology (lighting, detection, adaptive signage, . . . ) and a connection between mobility hubs, but also be a local source of green energy for the surrounding communities.

Within the project, the research was conducted into the optimum design of the solar panel roof with regards to aesthetic and energetic aspects, the environmental and social impact of the path, the smart technology and the segmentation of the (potential) customers/users and validation of their preferences, in order to fine tune the business model. This study concerns itself with the validation of the preferences of the (potential) customers/users, being Flemish commuters.

In order to identify the preference of different commuters regarding cycling infrastructure, an online survey was set-up. The survey was launched at four companies based in Flanders. The survey was sent out to all employees and completion was voluntary. The respondents were asked about socio-demographic characteristics with a focus on their commuting behavior.

Table 1 shows the population and the sample descriptives for the four companies.

The respondents were asked if they had either experience with cycling to work or if cycling was part of their commuting. Respondents were also asked which type of cycle they used for that commute at the time of the survey, being a conventional bicycle, a pedelec or a speed pedelec. Respondents using any type of cycle for their commute were categorized as 'cyclists' and the others as 'non-cyclists'. In making the distinction between 'conventional cyclists' ('CC') and 'EPAC users', the type of cycle they used was looked at. A non-electric

bicycle user was put in the 'conventional cyclist' group, a pedelec user or speed pedelec user was put in the 'EPAC user' group. This resulted in 578 non-cyclists and 302 cyclists, of which 165 ride a conventional bicycle and 137 ride an EPAC (112 pedelec and 25 speed pedelec). More descriptives for the different groups can be seen in Table 2.

**Table 1.** Descriptives of respondents.

|  | Company 1 | Company 2 | Company 3 | Company 4 |
|---|---|---|---|---|
| Employees (*n*) | ±120 | ±600 | ±350 | +3000 |
| Respondents (*n*) | 24 | 53 | 25 | 778 |
| Gender (ref: female), *n*, (%) | 7, (29%) | 11, (21%) | 9, (36%) | 545, (70%) |
| Age in years (Age in 2021) ($\bar{x}$), (SD) | 47, (11) | 42, (10) | 38, (8) | 41, (12) |
| Commute distance in km ($\bar{x}$), (SD) | 38, (27) | 31, (23) | 23, (22) | 16, (13) |

**Table 2.** Descriptives of different groups.

|  | Non Cyclists | Cyclists | EPAC Users | Conventional Cyclists |
|---|---|---|---|---|
|  |  | = | + |  |
| Respondents (*n*) | 578 | 302 | 137 | 165 |
| Gender (ref. female), *n*, (%) | 384, (66%) | 189, (63%) | 95, (69%) | 94, (57%) |
| Age in years (Age in 2021) (median), (SD) | 39, (12) | 41, (11) | 44, (12) | 40, (11) |
| Commute distance in km ($\bar{x}$), (SD) | 20, (15) | 14, (13) | 12, (9) | 16, (16) |

All respondents were asked to indicate to what extent certain aspects of cycling infrastructure would motivate/have a certain influence/be of importance to their choice of taking their (electric) bicycle to work using a five-point Likert-scale. The questions were grouped into four categories, being:

- General cycling path infrastructure;
- Infrastructure at cycling path stops;
- Aspects along cycling route;
- Signage along the cycling path.

The items of the different categories were based on a combination of items, from literature, insights from qualitative research performed before the launch of the survey and items determined by an expert panel group. The five items for '*General cycling path infrastructure*' are largely based on items found in [39,45], the six items for '*Aspects along the cycling route*'(distinction between route and path: a route consists of different parts, some are cycling paths) are based on insights from the focus groups and the panel group as well as items found in [45], the eight items for '*Infrastructure at cycling path stops*' are based upon the insights from the focus groups, panel group and items from [27,45], and the eight items for '*Signage along the cycling path*' are based on the insights from the panel group and on the items of [27,40]. Lastly, the respondents were asked, in line with the initial intention of the project, whether they would find each of the following useful: the construction of wide asphalted bicycle highways without a roof; with a roof and with a roof made up of photovoltaic (PV) panels providing green energy, which could be consumed locally. The question list is included in Appendix A as Table A1. The collected data were anonymized and analyzed by the researcher. The five-point Likert-scale scores were treated as an ordinal scale and for analysis in R they were transformed into numerical values, with '*Very unimportant*' as −2, '*Relative unimportant*' as −1, '*Neutral*' as 0, '*Relative important*' as 1 and '*Very important*' as 2.

## 3. Results

In this section, the results of the survey are described and discussed where appropriate. First, the general results are provided by showing the average of the answers for each item per group. In a second section, a factor analysis is used to examine to what extent

the items make up the provided constructs and whether there are statistically significant differences between the groups. The factor scores of the constructs are then used to check for correlations with the socio-demographics. The third section shows the statistically significant differences of the item level per group.

Table 3 shows the means, standard deviation and standardized factor loadings of those values per question or items per group. The factor loadings are described in Section 3.1. All items are positively formulated, except for items G1 to G5, which are negatively formulated. A negative score for these items should therefore be interpreted as follows: this item would have a bad influence to take a cycle to work. A positive score would imply a positive influence. When looking at the means and standard deviations of all items, it is striking here that only two items consistently over the four groups obtain a mean higher than one, being '*Quality of the cycle paths*' and '*Safe crossing points*'. '*Wide cycle paths*' also shows mean values above one, but only in the cyclists' groups. '*Secure bicycle parking*' and '*Smart traffic lights*' also score reasonably high in all groups. '*No cycle paths*', '*Poorly maintained cycle paths*' and '*Obstacles on the cycle paths*' score low in the non-cyclists' group and in the cyclists' group '*Rest areas*', '*Vending machines*' and '*Advertising signs*' score low. There are some items that are of non-interest for some groups by scoring a zero, the most noticeable being '*Seeing the sky above your head*' and '*Charging points*' for conventional cyclists and '*Cycle repair points*' for non-cyclists. Lastly, '*Wide asphalted bicycle highways without a roof*' score a mean higher than one for cyclists in contrast to '*Wide asphalted bicycle highways with a roof*' which leaves the cyclists rather indifferent, and is slightly more liked by non-cyclists. This changes, however, when that roof consists of PV panels providing green energy which can be consumed locally, where both non-cyclists and cyclists on average appear to be more in favor of the concept.

**Table 3.** Mean, standard deviations and standardized factor loadings (λ) of responses on items.

| Construct | Label | Items | Non-Cyclists | | | Cyclists | | EPAC Users | | Conventional Cyclists | |
|---|---|---|---|---|---|---|---|---|---|---|---|
| | | | Means (SD) | λ * | | Means (SD) | λ | Means (SD) | λ | Means (SD) | λ |
| **General cycle infrastructure** | | | | α = 0.92 | | | α = 0.89 | | α = 0.88 | | α = 0.89 |
| | G1 | Poorly maintained cycle paths | −1.0 (0.9) | 0.866 | | −0.8 (0.7) | 0.800 | −0.8 (0.8) | 0.769 | −0.8 (0.7) | 0.827 |
| | G2 | Narrow cycle paths | −0.7 (0.8) | 0.861 | | −0.6 (0.7) | 0.773 | −0.6 (0.8) | 0.750 | −0.6 (0.6) | 0.797 |
| | G3 | No cycle paths | −1.1 (0.9) | 0.829 | | −0.9 (0.9) | 0.789 | −0.9 (0.9) | 0.772 | −0.9 (0.9) | 0.811 |
| | G4 | Dirt on the cycle path | −0.6 (0.7) | 0.767 | | −0.6 (0.7) | 0.781 | −0.6 (0.7) | 0.807 | −0.5 (0.7) | 0.759 |
| | G5 | Obstacles on the cycle path | −0.9 (0.8) | 0.866 | | −0.8 (0.8) | 0.784 | −0.8 (0.8) | 0.802 | −0.8 (0.8) | 0.769 |
| **Aspects along cycling route** | | | | α = 0.77 | α = 0.90 | | α = 0.84 | | α = 0.84 | | α = 0.83 |
| | A1 | Stopping places | −0.3 (1.1) | 0.289 | / | −0.7 (1.0) | / | −0.8 (1.1) | / | −0.7 (1.0) | / |
| | A2 | Quality of cycle paths | 1.2 (1.0) | 0.902 | 0.910 | 1.4 (0.7) | 0.823 | 1.5 (0.7) | 0.846 | 1.4 (0.7) | 0.794 |
| | A3 | Safe crossing points | 1.3 (0.9) | 0.911 | 0.911 | 1.5 (0.7) | 0.797 | 1.6 (0.7) | 0.862 | 1.5 (0.7) | 0.736 |
| | A4 | Wide cycle paths | 0.9 (1.0) | 0.792 | 0.782 | 1.1 (0.8) | 0.769 | 1.1 (0.9) | 0.728 | 1.1 (0.7) | 0.830 |
| | A5 | Seeing the sky above your head | −0.1 (1.1) | 0.268 | / | 0.1 (0.9) | / | 0.3 (1.2) | / | 0.0 (1.1) | / |
| | A6 | Rain cover when cycling | 0.3 (1.0) | 0.391 | / | −0.1 (1.2) | / | −0.1 (1.2) | / | −0.1 (1.1) | / |
| **Infrastructure at stops** | | | | α = 0.86 | | | α = 0.84 | | α = 0.83 | | α = 0.84 |
| | I1 | Rest areas (benches and picnic areas) | −0.6 (1.1) | 0.664 | | −1.1 (1.0) | 0.572 | −1.2 (1.0) | 0.572 | −1.0 (1.0) | 0.573 |
| | I2 | Secure bicycle parking | 0.9 (1.3) | 0.559 | | 0.9 (1.3) | 0.425 | 0.8 (1.4) | 0.443 | 1.0 (1.3) | 0.420 |
| | I3 | Charging points for EPACs | 0.5 (1.3) | 0.690 | | 0.2 (1.4) | 0.652 | 0.5 (1.4) | 0.607 | 0.0 (1.3) | 0.746 |
| | I4 | Mobility hubs | 0.2 (1.1) | 0.715 | | 0.0 (1.2) | 0.713 | 0.1 (1.2) | 0.744 | −0.1 (1.2) | 0.709 |
| | I5 | Vending machines | −0.8 (1.1) | 0.617 | | −1.1 (1.0) | 0.608 | −1.2 (1.0) | 0.595 | −1.0 (1.0) | 0.609 |
| | I6 | Parking facilities for cars at starting points | 0.4 (1.3) | 0.629 | | −0.4 (1.4) | 0.654 | −0.5 (1.3) | 0.717 | −0.3 (1.5) | 0.617 |
| | I7 | Cycle repair points | 0.0 (1.1) | 0.680 | | 0.0 (1.2) | 0.685 | −0.2 (1.2) | 0.639 | 0.1 (1.2) | 0.715 |
| | I8 | Sufficient waiting room | −0.2 (1.1) | 0.726 | | −0.6 (1.1) | 0.753 | −0.6 (1.1) | 0.707 | −0.6 (1.1) | 0.776 |
| **Signage along the cycling infrastructure** | | | | α = 0.85 | | | α = 0.84 | | α = 0.82 | | α = 0.86 |
| | S1 | Smart traffic lights | 0.7 (0.8) | 0.714 | | 0.9 (0.9) | 0.680 | 0.9 (1.0) | 0.624 | 0.8 (0.9) | 0.743 |
| | S2 | Speed signs | 0.2 (0.8) | 0.684 | | 0.0 (0.9) | 0.754 | 0.0 (0.9) | 0.762 | 0.1 (0.9) | 0.745 |
| | S3 | Smart traffic signs | 0.4 (0.8) | 0.840 | | 0.4 (0.9) | 0.861 | 0.4 (1.0) | 0.839 | 0.4 (0.9) | 0.884 |
| | S4 | Dynamic lane signs | 0.4 (0.8) | 0.868 | | 0.3 (0.9) | 0.862 | 0.3 (1.0) | 0.854 | 0.4 (0.8) | 0.877 |
| | S5 | Variable traffic signs | 0.3 (0.8) | 0.899 | | 0.3 (0.9) | 0.828 | 0.2 (0.9) | 0.780 | 0.4 (0.8) | 0.869 |
| | S6 | Advertising signs | −0.8 (0.9) | 0.089 | | −1.0 (0.9) | 0.109 | −1.1 (0.9) | 0.073 | −0.9 (0.9) | 0.121 |
| | S7 | Variable message signs | 0.1 (0.8) | 0.635 | | 0.0 (0.9) | 0.519 | 0.0 (1.0) | 0.484 | 0.1 (0.9) | 0.537 |
| | S8 | Overtaking lanes for fast cyclists | 0.5 (0.9) | 0.567 | | 0.7 (1.0) | 0.466 | 0.7 (1.1) | 0.419 | 0.7 (0.9) | 0.509 |
| **Wide asphalted bicycle highways** | | | | | | | | | | | |
| | P1 | Without a roof | 0.8 (0.9) | | | 1.1 (0.8) | | 1.2 (0.8) | | 1.1 (0.8) | |
| | P2 | With a roof | 0.5 (0.9) | | | 0.2 (1.2) | | 0.2 (1.2) | | 0.2 (1.2) | |
| | P3 | With a roof made up of PV panels providing green energy which could be consumed locally | 0.9 (1.0) | | | 0.8 (1.1) | | 0.8 (1.1) | | 0.8 (1.1) | |

* λ: standardized factor loading.

### 3.1. Hypotheses Testing on Construct Level

The questions, or rather, items, have a certain weight on the overarching categories or constructs. To evaluate that weight for each group of respondents, a confirmatory analysis

was used. First, it is advised to check the internal consistency of the construct by calculating the Cronbach Alpha. If the Cronbach Alpha value is above the threshold value of 0.7, the items measure the construct effectively. The standardized factor loadings symbolize the weight of every item on the construct. The alphas for each construct as well as the standardized factor loadings are shown in Table 3. Note that the alphas and factor scores for the three items of '*Wide asphalted bicycle highways*' are not included in the table, as they can neither be grouped as a construct, nor is there a plausible theoretical foundation. Still, these items have significance within the scope of the project and will therefore be discussed in Section 3.2. In Table 3, it can be noticed that all alphas are above the threshold value but the value for '*Aspects along cycling route*' for the cyclists, conventional cyclists and EPAC users was calculated without items A1, A5 and A6. Those times were excluded to achieve a Cronbach Alpha above the threshold value. The exclusion was not necessary for the non-cyclists, as can be observed in the table, but nevertheless, was executed to compare the non-cyclists with the cyclists. For exhaustiveness reasons, the Cronbach Alpha and the factor loadings for the non-cyclists were included with and without the exclusion of A1, A5 and A6. Furthermore, it can be noticed that item S6 '*Advertising signs*' has overall very low loadings, indicating that S6 does not motivate taking a cycle to work for all respondents, in contrast with the other items.

To now determine statistically significant differences between the groups at a construct level, factor scores were determined to serve as input for a Mann–Whitney $U$-test. The factor scores for the four constructs were calculated with the lavInspect function of the lavaan-package [46] in R for each category of respondent (i.e., non-cyclist, cyclist, EPAC user, conventional cyclists). Table 4 shows the $p$-values as a result of the Mann–Whitney $U$-tests, where a threshold value of $p = 0.05$ is taken to reject the null hypothesis that both groups are equal. The alternative hypothesis is evidently that the groups are statistically significantly different and therefore indicate that the requirements for the two groups are different.

**Table 4.** Hypothesis testing on construct level.

| Constructs | Cyclists vs. Non-Cyclists ($p$-Value) | Null Hypothesis Failed to Reject/Rejected | EPAC Users vs. CC ($p$-Value) | Null Hypothesis Failed to Reject/Rejected |
|---|---|---|---|---|
| General | 0.8445 | Failed to reject | 0.4871 | Failed to reject |
| Aspects | 0.0001674 | Rejected | 0.7233 | Failed to reject |
| Stops | 0.7496 | Failed to reject | 0.8385 | Failed to reject |
| Signage | 0.1651 | Failed to reject | 0.7178 | Failed to reject |

It can be concluded that for cyclists and non-cyclists the requirements only differ for the construct '*Aspects along the cycling route*'. There are no significant differences on the construct level between the EPAC users and conventional cyclists.

The factor scores of the constructs are now used to check for correlations with the socio-demographics, using the Spearman Rho correlation test, as the data are non-parametric. As shown in Table 5, no correlations were found between the factors and the socio-demographic information (i.e., gender (1 = male, 2 = female), age and commute distance). This shows that there are neither positive nor negative relations between the socio-demographics and the factors.

**Table 5.** Spearman correlations among socio-demographics and the factors for each group.

| | Non-Cyclists | | | Cyclists | | | EPAC Users | | | Conventional Cyclists | | |
|---|---|---|---|---|---|---|---|---|---|---|---|---|
| | **1** | **2** | **3** | **1** | **2** | **3** | **1** | **2** | **3** | **1** | **2** | **3** |
| 1. Gender | - | | | - | | | - | | | - | | |
| 2. Age | 0.11 | - | | 0.08 | - | | −0.02 | - | | 0.19 | - | |
| 3. Commute distance | −0.09 | −0.07 | - | −0.1 | −0.02 | - | −0.15 | −0.06 | - | −0.05 | 0 | - |
| 4. General | 0.01 | 0.18 | 0.11 | 0.06 | −0.08 | −0.13 | 0.13 | −0.02 | −0.19 | −0.01 | −0.17 | −0.09 |
| 5. Aspects | 0.13 | −0.13 | −0.06 | 0.07 | −0.12 | −0.06 | 0.13 | −0.06 | −0.15 | −0.01 | −0.14 | −0.04 |
| 6. Stops | 0.19 | −0.08 | 0.1 | 0.07 | 0.07 | 0.13 | 0.06 | 0.04 | 0.13 | 0.06 | 0.08 | 0.12 |
| 7. Signage | −0.07 | −0.16 | 0.06 | −0.07 | 0.01 | 0.03 | −0.14 | 0.05 | 0.09 | −0.01 | −0.05 | −0.02 |

*3.2. Hypothesis Testing on Item Level*

In the previous section, the factor loadings were calculated, and it was determined whether statistically significant differences between the factor scores of the constructs were present for the two groups (i.e., non-cyclists vs. cyclists and EPAC users vs. conventional cyclists). The following sections look at the statistical differences between the groups on an item level. First, a general overview is given and secondly, the statistical differences for both groups '*cyclists* vs. *non-cyclists*' and '*EPAC users* vs. *conventional cyclists*' are further described.

To calculate the statistically significant differences between two independent groups of non-parametric data with ordinal values, the Mann–Whitney U-test is used. When there are multiple tests being undertaken, there is a chance for a type I error, i.e., rejecting a null hypothesis that should not be rejected, because the *p*-value is by chance beneath the threshold. To avoid multiple testing errors, the Bonferroni [47] method is used, whereby the threshold *p*-value is divided by the number of tests conducted on one dataset. In this case, the threshold value is:

$$p_{Bonferroni} = \frac{0.05}{30} = 0.001667 \tag{1}$$

The *p*-values, as a result of the Mann–Whitney $U$-tests, are shown in Table 6. The values that are below 0.05 and 0.01 are indicated with '<0.05' and '<0.01', respectively, and the values below the $p_{Bonferroni}$ are indicated with '<0.001667'. It can be noticed that there are 12 items with statistically significant differences within the 'cyclists vs. non-cyclists' group and there is only one within the 'conventional cyclists vs. EPAC users' group.

**Table 6.** Mann–Whitney U-test *p*-values on item level.

| Construct | Label | Item | Cyclists vs. Non-Cyclists | EPAC Users vs. CC |
|---|---|---|---|---|
| | | | Mann–Whitney $U$-Test | |
| **General cycle infrastructure** | | | *p*-values | |
| | G1 | Poorly maintained cycle paths | <0.05 | 0.49 |
| | G2 | Narrow cycle paths | 0.05 | 0.85 |
| | G3 | No cycle paths | <0.01 | 0.93 |
| | G4 | Dirt on the cycle path | 0.24 | 0.58 |
| | G5 | Obstacles on the cycle path | 0.27 | 0.85 |
| **Aspects along cycling route** | | | | |
| | A1 | Stopping places | <0.001667 | 0.18 |
| | A2 | Quality of cycle paths | <0.01 | <0.05 |
| | A3 | Safe crossing points | <0.01 | <0.01 |
| | A4 | Wide cycle paths | <0.01 | 0.32 |
| | A5 | Seeing the sky above your head | <0.001667 | 0.11 |
| | A6 | Rain cover when cycling | <0.001667 | 0.74 |
| **Infrastructure at stops** | | | | |
| | I1 | Rest areas (benches and picnic areas) | <0.001667 | 0.10 |
| | I2 | Secure bicycle parking | 0.77 | 0.55 |
| | I3 | Charging points for EPACs | <0.05 | <0.001667 |
| | I4 | Mobility hubs | <0.01 | 0.16 |
| | I5 | Vending machines | <0.001667 | 0.24 |
| | I6 | Parking facilities for cars at starting points | <0.001667 | 0.16 |
| | I7 | Cycle repair points | 0.56 | 0.09 |
| | I8 | Sufficient waiting room | <0.001667 | 0.68 |

**Table 6.** *Cont.*

| Construct | Label | Item | Cyclists vs. Non-Cyclists | EPAC Users vs. CC |
|---|---|---|---|---|
| | | | Mann–Whitney *U*-Test | |
| **Signage along the cycling infrastructure** | | | | |
| | S1 | Smart traffic lights | <0.001667 | 0.53 |
| | S2 | Speed signs | <0.05 | 0.24 |
| | S3 | Smart traffic signs | 0.67 | 0.38 |
| | S4 | Dynamic lane signs | 0.57 | 0.18 |
| | S5 | Variable traffic signs | 0.54 | 0.09 |
| | S6 | Advertising signs | <0.001667 | <0.05 |
| | S7 | Variable message signs | 0.98 | 0.30 |
| | S8 | Overtaking lanes for fast cyclists | <0.001667 | 0.74 |
| **Wide asphalted bicycle highways** | | | | |
| | P1 | Without a roof | <0.001667 | 0.19 |
| | P2 | With a roof | <0.001667 | 0.87 |
| | P3 | With a PV panel roof | 0.662 | 0.89 |

### 3.2.1. Differences between Conventional Cyclists and EPAC Users

It can be concluded that EPAC users and conventional cyclists in this sample do not differ in preferences towards cycle infrastructure, expect for one item. It might not be surprising that for EPAC users, '*Charging points at stops*' are much more important than for conventional cyclists. Table 6 also shows the items that have a *p*-value less than 0.05, which in this case are '*Quality of the cycle paths*' (A2), '*Safe crossing points*' (A3) and '*Advertising signs*' (S6). Table 3 shows that for EPAC users, A2 and A3 are slightly more important than for conventional cyclists and S6 is much more negatively rated by EPAC users than it is by conventional cyclists. The conservative method of Bonferroni correction excludes these items from analysis, and could potentially ignore these items as significantly different. More research on differences between EPAC users and conventional cyclists for these items would be needed to determine this, especially because the aspects A2 and A3 could be of interest for EPAC users to maintain their higher average speeds.

### 3.2.2. Differences between Cyclists and Non-Cyclists

The differences between cyclists and non-cyclists in this sample can be observed in Table 6. The items with a significant statistical difference in the construct '*Aspects along the cycling route*' are '*Stopping places*' (A1), '*Seeing the sky above your head*' (A5) and '*Rain cover when cycling*' (A6). The first item, A1, is seen as more negative by cyclists, than by non-cyclists. A5 is seen as quite neutral by both groups, but slightly more positive by the cyclists. A6 is then again seen as more positive by non-cyclists, compared to the opinion of the cyclists. Within the construct '*Infrastructure at stops*', the cyclists in this sample find '*Rest areas*' (I1) and '*Vending machines*' (I5) less important than non-cyclists, and the non-cyclists in turn find '*Parking facilities for the car*' (I6) and '*Waiting areas for colleagues*' (I8) more important than taking their cycle to work. Within the construct '*Signage along the cycling infrastructure*' are '*Smart traffic lights*' (S1), '*Advertising signs*' (S6) and '*Overtaking lanes for fast cycles*' (S8). Cyclists find smart traffic lights and overtaking lanes for fast cyclists more important than non-cyclists. Advertising signs are less important for cyclists than they are for non-cyclists.

For both items P1 and P2, a statistically significant difference is found between the answers of the cyclists and the non-cyclists, as seen in Table 6. Cyclists find the construction of '*Wide asphalted bicycle highways without a roof*' (P1) a lot more useful than non-cyclists. Non-cyclists in turn find '*Wide asphalted bicycle highways with a roof*' (P2) a bit more useful than cyclists, however, much less than they did the '*Wide asphalted bicycle highways without a roof*'. The third item, P3, did not have a statistical difference. Both non-cyclists and cyclists, however, find on average the construction of '*Wide asphalted bicycle highways with a roof made up of PV panels providing green energy which could be consumed locally*' much more

useful in comparison to a regular roof. This could be of course influenced by a social desirability bias.

The following items also have a *p*-value below 0.05 when looking at differences between cyclists and non-cyclists: 'Poorly maintained cycle paths', 'No cycle paths', 'Quality of cycle paths', 'Safe crossing points', 'Wide cycle paths', Charging points', 'Mobility hubs' and 'Speed signs'. As mentioned in Section 3.2.2, more research on the differences between cyclists and non-cyclists for these items would be needed to determine if the Bonferroni method is falsely excluding these items.

## 4. Discussion and Conclusions

This paper presents the results of a survey performed in the context of the SEBP project on preferences towards cycle infrastructure and discusses the differences of preference between different groups: cyclists vs. non-cyclists and conventional cyclists vs. EPAC users. The survey was sent out to four companies in Flanders, Belgium, with a total of 880 respondents. This paper analyses the 27 items categorized into four constructs: '*General cycle infrastructure*', '*Aspects along the cycle route*', '*Infrastructure at stops*' and '*Signage at the cycle infrastructure*' and three items specifically related to the SEBP project. The study wanted to identify what cyclists want in terms of cycling infrastructure and determine whether this differs from non-cyclists. In addition, it was examined whether there is a difference within the group of cyclists between EPAC users and conventional cyclists.

The literature demonstrates that the lack of appropriate cycling infrastructure is one of the main barriers keeping Flemish commuters from cycling to work [31,34–36]. Providing appropriate cycling infrastructure, along with other measures [34–36], will increase cycling [5,16–24]. The results of this survey support this when looking at the responses; both cyclists and non-cyclists see the presence of safe crossing points and the quality of the cycle paths as the most important aspects in cycle infrastructure. Wide cycle paths, secure bicycle parking and smart traffic lights also score high in both groups. Moreover, the negative scoring on items such as '*No cycle paths*' and '*Poorly maintained cycle paths*' shows that both groups, but especially non-cyclists, express a need for appropriate cycling infrastructure to cycle to work. Other suggested innovations, for example smart and dynamic signage, were considered quite neutral in convincing all groups to take a cycle to work. With regard to the differences between the needs of the different groups, the differences at the construct level were first evaluated and correlations with the socio-demographics were checked. Secondly, the differences on the item level were calculated. The evaluation on the construct level did not find any correlation between the constructs and the socio-demographics nor did it find statistical differences between the groups, except for a difference between the cyclists and non-cyclists for the construct '*Aspects along the cycling route*'.

To the authors' knowledge, no sources report on the differences in needs between EPAC users and conventional cyclists towards cycling infrastructure. Their needs are apparently assumed similar, despite definite distinctions in characteristics of the cycles (i.e., speed, weight, electrical components, price, speed differences) [30,31,41–43]. The results of this study demonstrate that this assumption appears to be correct, with the only, maybe unsurprising, statistically significant difference, the importance attributed to '*Charging points for EPACs*'.

Regarding the needs of cyclists and non-cyclists, 12 statistically significant differences were found. Non-cyclists find '*Stopping places*', '*Rest areas*', '*Vending machines*', '*Sufficient waiting room*' and '*Advertising signs*' unimportant, but significantly less so than cyclists. Non-cyclists find '*Parking facilities for cars at starting points*' and '*Rain cover when cycling*' important in contrast to cyclists, who find both rather unimportant. The preference of non-cyclists for protection from the rain is reflected in the answers to the question of whether the construction of '*Wide asphalted bicycle highways with a roof*' would be useful. The answers of non-cyclists are statistically significantly different than those of cyclists, finding it more useful, which is a finding that is confirmed by [24]. Cyclists find '*Smart traffic lights*' and

'*Overtaking lanes for fast cyclists*' significantly more important than non-cyclists. They are also a bit more positive towards '*Seeing the sky above your head*' while cycling, which is reflected in their answer on the usefulness of constructing '*Wide asphalted bicycle highways without a roof*', which cyclists find statistically significantly more important compared to non-cyclists. Finally, it should be mentioned that both cyclists and non-cyclists find the construction of '*Wide asphalted bicycle highways with a roof made up of PV panels providing green energy which would be consumed locally*' more useful than a roof without such PV panels.

*Implications for Policy and Future Research*

Policy makers should firstly focus on providing wide quality cycle paths with safe crossing points, smart traffic lights and secure bicycle parking, as these are the most important aspects in terms of cycle infrastructure for both cyclists and non-cyclists. Secondly, they should cater both to the needs of cyclist and non-cyclists, as the efforts of both are needed to reach the European Green Deal objectives. Apart from the charging points for EPACs along the cycle route, no statistically significant differences between the preferences of conventional cyclists compared to EPAC users were found. This means an overhaul of existing infrastructure is not necessary and the guidelines of the European Commission [38] and local authorities [37] can be followed. More research, however, will be needed to check whether the Bonferroni correction does not falsely reject items.

This paper adds to the literature, as it covers the different needs non-, conventional and electric cyclists have for cycling infrastructure. Further research is needed, however, to validate the current results with a larger and more representative sample size for the Belgian commuters and to check whether the Bonferroni does not falsely reject items. How and where best to implement these charging points is a subject not dealt with within the scope of this paper, but deserves further research. The study was conducted in Flanders, Belgium, a country with a high modal share [48]. Therefore, the transferability of the results to other region or countries should be further investigated. Finally, a longitudinal study, with pre- and post-measurements of the opinions and behavior of both cyclists and non-cyclists using a newly built cycle path with a roof cover and the above-mentioned infrastructure could support the theoretical nature of the present study and an analysis via structural equation modeling could further validate its findings.

**Author Contributions:** Conceptualization, N.V.d.S., J.C., L.V. and B.d.G.; methodology, N.V.d.S., L.V. and B.d.G.; formal analysis, N.V.d.S.; investigation, N.V.d.S.; resources, J.C. and L.V.; data curation, N.V.d.S.; writing—original draft preparation, N.V.d.S.; writing—review and editing, J.C., L.V. and B.d.G.; visualization, N.V.d.S.; supervision, J.C. and L.V.; project administration, N.V.d.S.; funding acquisition, J.C. and L.V. All authors have read and agreed to the published version of the manuscript.

**Funding:** This research was funded by Flux50 with funding number 2020/1603 AIO-FLUX50-CON.

**Institutional Review Board Statement:** Ethical review and approval were waived for this study in agreement with the internal guidelines of the VUB Research & Data management department, as this study does not deal with vulnerable groups, children/minors or people who cannot give informed consent.

**Informed Consent Statement:** Informed consent was obtained from all subjects involved in the study.

**Data Availability Statement:** Not applicable.

**Acknowledgments:** We acknowledge the financial support of FLUX50, the project setting and partners (BAM Belgium, VUB, KU Leuven, Menapy, ABB Belgium and AE) of Smart Energy Bike Path (SEBP) project where the survey is part of and thank all the companies participating in the survey; BAM Belgium, ABB Belgium, AE, UZ Brussel.

**Conflicts of Interest:** The authors declare no conflict of interest. The funders had no role in the design of the study; in the collection, analyses, or interpretation of data; in the writing of the manuscript, or in the decision to publish the results.

## Appendix A

**Table A1.** Questions of survey with Likert scale.

| Labels | Questions | Likert Scale |
|---|---|---|
| *To what extent do the following cycling infrastructure factors influence your decision to cycle to work in the current situation?* | | |
| G1 | Poorly maintained cycle paths | Very bad influence ($-2$) |
| G2 | Narrow cycle paths | Bad influence ($-1$) |
| G3 | No cycle paths | No influence (0) |
| G4 | Dirt on the cycle path | Positive influence (1) |
| G5 | Obstacles on the cycle path | Very positive influence (2) |
| *To what extent are the following aspects along your cycle route important if you would cycle to work?* | | |
| A1 | Stopping places | Very unimportant ($-2$) |
| A2 | Quality of cycle paths | Unimportant ($-1$) |
| A3 | Safe crossing points | Neither important nor |
| A4 | Wide cycle paths | unimportance (0) |
| A5 | Seeing the sky above your head | Important (1) |
| A6 | Rain cover when cycling | Very important (2) |
| *To what extent are the following aspects at a stopping place important if you take your cycle to work?* | | |
| I1 | Rest areas (benches, picnic areas) | |
| I2 | Secure bicycle parking | Very unimportant ($-2$) |
| I3 | Charging points for EPACs | Unimportant ($-1$) |
| I4 | Mobility hubs | Neither important nor |
| I5 | Vending machines | unimportance (0) |
| I6 | Parking facilities for cars at starting points | Important (1) |
| I7 | Cycle repair points | Very important (2) |
| I8 | Sufficient waiting room | |
| *To what extent do the following signage elements on your cycle route motivate you to take your cycle to work?* | | |
| S1 | Smart traffic lights | |
| S2 | Speed signs | Totally not motivating ($-2$) |
| S3 | Smart traffic signs | Not motivating ($-1$) |
| S4 | Dynamic lane signs | Neither motivating nor |
| S5 | Variable traffic signs | motivating (0) |
| S6 | Advertising signs | Motivating (1) |
| S7 | Variable message signs | Very motivating (2) |
| S8 | Overtaking lanes for fast cyclists | |
| *Indicate to what extent you find the following useful:*<br>*The construction of wide asphalted bicycle highways* | | |
| P1 | Without a roof | Very useless ($-2$) |
| P2 | With a roof | Useless ($-1$) |
| | With a roof made up of PV panels | Neither useless, nor |
| | providing green energy which could be | useful (0) |
| P3 | consumed locally | Useful (1) |
| | | Very useful (2) |

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
