# Peer review of "Cycling Infrastructure for All EPACs Included?"

_wevj, doi:10.3390/wevj13050074_

Round 1
Reviewer 1 Report
The manuscript concerns an important topic and the idea and findings of this study have merits. The background, literature, gaps and research questions are clearly formulated. I have a few points that the authors need to address prior to publication:
- Table 1: The distribution of female respondents for Company 1 looks ambiguous. Please check it.
- Table 1: I suggest to change the median values with mean values for the Age.
- Table 2: How did you deal with cases, where the cyclists were both the Conventional and EPAC users? Did you ask them during your survey if they use both types of bikes?
- The results provided in Table 3 looks very confusing. First of all, I recommend to change the mean values at least up to 2 decimal places. In addition, what for instance the mean value of -1.0 for G1 and non-cyclist represent? It looks like that "Poorly maintained cycle paths" are relatively unimportant to the non-cyclists? I suggest to reformulate the questions (or their coding) in a way to reduce these confusions.
- Adding to my comment#4, I think it is important to state the exact questions that were asked for the items given in Table 3. An alternative could be to provide the questionnaire as an appendix.
- Table 5: it looks like that there are no major differences between the tested groups except for the "Aspects along cycling route". Did you try ordinal regression instead?
- I am not convinced of the choice of using Fisher exact tests. I think they should use the same non-parametric tests (Mann-Whitney U or Kolmogorov-Smirnov). Otherwise, I recommend using Ordinal regressions.
- Why did you not consider gender, age and other contextual factors (such as, commute distance) in the analyses? I think including these factors could enhance the overall results.
- The policy implications aspect of the paper can be improved.
Reviewer 2 Report
The authors present the findings of a survey undertaken amongst commuters in Flanders, Brussels. Overall, the manuscript is clearly written, and the findings are clear.
I thought the results were very repetitive, particularly the key findings which are presented in Table 3, which are then tested for statistical significance in Table 6 and then plotted as distributions in Figures 2 onwards. Much of these findings could be consolidated into a single table, which would significantly shorten the manuscript. Most of the graphs presented in the report are unnecessary and offer very limited value to the reader, particularly the box plots in Figure 1.
There is scope to extend the factor analysis, at least the authors could explore correlations between factors and demographics. Even more interesting could be investigating the interactions between factors using structural equation modelling.
The conclusion would benefit from extensions where the authors discuss the implications of the findings in terms of the research questions. The authors also need to link their findings to the academic literature. Currently there are no references in the conclusion or the results/ discussion section.
Some minor comments:
Line 26 – include reference
Line 380 – fix heading
Check figure captions and delete unnecessary figures.
Round 2
Reviewer 1 Report
The authors have properly addressed all of my comments. I can see that the analyses are now improved and other typos are resolved. I do not have any further remarks
Author Response
Dear reviewer,
We would like to thank you for the review and the comments on our manuscript.
They have contributed to make this manuscript a better whole.
Kind regards,
The authors
Reviewer 2 Report
The authors have made the suggested amendments to the manuscript and in my opinion the research is suitable for publication.
A minor note to fix the hyperlinks in the text as error messages are being displayed.
Author Response
Dear reviewer,
We would like to thank you for the review and the comments on our manuscript.
They have contributed to make this manuscript a better whole.
Hyperlinks should be fixed in current version of the manuscript.
Kind regards,